# An extrapolation algorithm for estimating river bed grain size distributions across basins

Jordan T. Gilbert[1,2]

[1]Department of Geosciences, University of Montana, Missoula, MT 59812-1296, United States

[2]Department of Watershed Sciences, Utah State University, Logan, UT 84322, United States

*Correspondence to:* Jordan Gilbert (jordan.gilbert@usu.edu)

**Abstract.** Values representing grain size distributions of stream reaches are essential for estimating sediment transport at the reach scale. Various modeling frameworks exist that attempt to simulate reach-scale sediment transport across entire drainage basins to characterize sediment dynamics at a watershed scale. Such frameworks require estimates of grain size at each reach. Because obtaining direct measurements at this scale is impractical and logistically difficult, methods to estimate or extrapolate grain size measurements are needed, however, few currently

exist. Here I present an extrapolation algorithm that uses one or more pebble counts to extrapolate full grain size distributions to each reach of a drainage network. In addition to the pebble count measurements, the tool requires a stream network geospatial feature class, attributed with values for reach-averaged slope and some consistent measure of relative flow magnitude (or a proxy for flow). I tested the tool in a set of sub-watersheds in the Bitterroot River basin of western Montana, US, with varying valley morphologies, and compared predictions to measurements

at 16 sites. When using multiple measurements for calibration, mean absolute percent error averaged 5.8% of the measured grain sizes in the phi scale. When using a single measurement for calibration, error averaged 8.4%.

## 1 Introduction

A characterization of the bed surface grain size distribution (GSD) of rivers is essential for applying sediment transport formulae to estimate transport rates and total yield. In cases where the bed can be assumed to be equally

mobile, a single characteristic size (e.g. median grain size, $D_{50}$) is often used to represent the bed material (e.g.,

Meyer-Peter & Müller, 1948). In contrast, to apply fractional transport equations (e.g., Parker, 1990; Wilcock & Crowe, 2003), a GSD is needed. GSD data can be tedious and time consuming to collect, however, and spatial coverage within a given area can be limited by accessibility to channels. Application is, therefore, generally limited in spatial extent unless GSDs can be estimated for locations without direct measurements.


Two general methods are typically used for quantifying GSDs: pebble counts and photo sieving. Pebble counts (often referred to as Wolman pebble counts after Wolman (1954)) are a simple method of characterizing bed GSDs by randomly gathering particles from the bed surface within a defined extent, and measuring the b-axis diameter of each. Sampling bias is common as larger particles are easier to see and are typically touched first as they protrude

from the bed more, especially in poorly sorted mixtures (Leopold, 1970; Wohl et al., 1996).  To avoid sampling bias, protocols have been recommended when performing pebble counts.  For example, a "random walk" method is often used, with the sampler moving randomly across the defined area, identifying the particle at the point of the foot at each step and *then* retrieving it, rather than reaching down and grabbing the first particle touched. Kondolf (1997) recommends performing individual pebble counts for unique populations (geomorphic units) within a reach and

taking the area-weighted average to characterize the whole reach. Another method uses a grid system, identifying the particle directly beneath each intersection and retrieving it as part of the count (Bunte & Abt, 2001). Typical pebble counts consist of 100 particles; fewer may be needed to characterize relatively homogenous beds, and more may be needed in poorly sorted beds with a wide GSD. Additionally, multiple pebble counts can be performed for a given location by a single person or multiple individuals, and the results used to quantify uncertainty in the values

used (e.g., $D_{50}$). Despite the drawbacks of this method related to sampling bias, it is easy to implement, provides good characterizations of river beds, and is the most widely used method.

Photo sieving is a methodology in which image processing algorithms are used to identify individual grains in photographs and calculate their diameter (Ibbeken & Schleyer, 1986). The advantages of this method are that rather

than a random sample, all identifiable particles can be used to generate the GSD, and it can be applied over large areas wherever photographs are available. These algorithms have been improved upon with advances in computer vision technology (e.g., Buscombe, 2008), including even broader application with the advent of drone-based

surveys, which can produce orthoimagery of very large areas of river bed (e.g., Purinton & Bookhagen, 2019).

Drawbacks of this method are that it can only be effectively applied where grains are exposed and easily identifiable

in images, and that partially buried or obscured grains that are still part of the surface layer cannot be accurately

measured. This methodology, therefore, cannot be applied to inundated or densely vegetated areas. Where

applicable, photo sieving produces results comparable to pebble counts (Strom et al., 2010). Where it is appropriate

to use, photo sieving can characterize grain size over larger spatial scales than pebble counts, however, like pebble

counts, it is an impractical method for quantifying GSDs across entire drainage networks.


Several additional methods exist for characterizing bed GSDs. For very fine sediments, laser diffraction can be used

to measure particle size (Blott et al., 2004; Eshel et al., 2004). For fine beds (silt to gravel), samples can be taken in

the field that are separated into different size fractions in the lab using sieves or settling tubes (Komar & Kui, 1984).

When available, very high-resolution topography of the bed (e.g., from terrestrial or mobile laser scanners or

structure-from-motion photogrammetry) to estimate median grain sizes based on relationships between grain size

and roughness (Pearson et al., 2017; Williams et al., 2019).

Controls on grain size include various factors relating to sediment production processes, as well as hydraulic sorting

of bed material. For example, lithology exerts control on sediment delivery rates to channels, as well as the size of

that sediment (Mueller et al., 2016). The size of individual grains produced from a rock layer is also a function of

fracture spacing within the rock (Neely & DiBiase, 2020; Vericat et al., 2021). When sediment is delivered to

channels from hillslope erosion processes, hydraulic sorting then exerts control on what sizes are present in the bed

at a given reach (Snelder et al., 2011). As a result, variables often associated with sediment transport (i.e., channel

gradient and flow magnitude) also correlate with bed GSD (Cohen et al., 2022).


Various sediment models seek to simulate transport across drainage networks in a spatially explicit manner (e.g.,

Czuba, 2018; Gilbert & Wilcox, 2020; Schmitt et al., 2016), and thus require a specification of grain size at each

spatially distinct location in the modeling domain. These models have traditionally used relatively simple

approaches, including applying the same distribution to the entire network, or correlating a median grain size with topographic and hydrologic information. A systematic method for estimating full distributions across drainage networks would thus be a valuable contribution for improving existing sediment transport and connectivity models.

Several approaches have been used to estimate grain size across large spatial extents. Jha et al. (2022) assessed $D_{50}$ at over 2,400 locations from five different sources and used inverse distance weighted interpolation to estimate $D_{50}$ regionally between sample points. This provides broad regional patterns, but does not provide information in specific catchments or reaches. Abeshu et al. (2022) attributed $D_{50}$ information from 2,577 United States Geologic Survey (USGS) gages to corresponding reaches in the National Hydrography Dataset (NHD) and used machine learning to develop a model based on the attributes available in the in NHDPlus dataset, finding that slope and distance between the reach and the stream's mouth were the most important channel characteristic predictors of grain size. Similarly, Snelder et al. (2011) used random forest models to assess data from 677 stream reaches in France with grain size measurements and found that the model incorporating variables related to hydraulic sorting (e.g., slope and precipitation) best predicted $D_{50}$.

Other approaches for estimating grain size across large spatial scales have used remote sensing-based methods. For example, Snyder et al. (2013) used digital elevation models to predict grain size. Their approach requires spatially distributed estimates of Manning's n roughness values, and is based upon the assumption that critical Shields stress ($\tau^*_c$) is 0.04 and that it is approximately equal to the Shields stress of a 2-year recurrence-interval flood. Similarly, models based on Shields stress (Buffington et al., 2004) and stream power (Gorman et al., 2011), calibrated empirically with both GIS and field data, have been used to develop predictive grain size models. However, all of the approaches described here estimate only the $D_{50}$, and no method currently exists that predicts full GSDs across drainage networks, despite the fact that such information would be useful for a variety of purposes.

In this paper, I introduce a method for extrapolating grain size distributions for coarse-bedded rivers (sand and coarser) throughout a drainage network based on one or more measurements (e.g., obtained using pebble counts or

photo sieving). This method, similar to previous approaches, has the advantage of being calibrated based on actual measurements for a basin, while utilizing remote sensing data to obtain topographic data. It also predicts a full grain size distribution for each reach, rather than just $D_{50}$, allowing for application of fractional sediment transport functions. It also essentially estimates grain sizes as a function of hydraulics. Therefore, because hillslope sediment delivery is an additional control on grain sizes present within a reach (Hassan et al., 2019), this approach is best suited to alluvial river systems, systems with minimal coarse sediment inputs from hillslopes, or systems where the magnitude of hillslope sediment production is not highly spatially variable.

## 2 Methods

The grain size tool presented here can use a single measurement, or multiple measurements of GSD to extrapolate estimates of $D_{16}$ (the particle size for which 16% of the bed is finer), $D_{50}$, and $D_{84}$ (the particle size for which 84% of the bed is finer), along with an uncertainty range for each to every segment or reach of a geospatial drainage network layer. The tool then generates an estimate of full GSDs for each reach (i.e., the fraction of each half-phi size class in the bed; this information is stored in a '.json' file). Because relative size differences are more meaningful than absolute size differences in GSDs (i.e., a difference of 1 mm from a 1 mm to a 2mm particle has much greater implications for transport than from a 150 to 151 mm particle), the logarithmic transformation of particle sizes from standard units (mm) to phi units is often used (Tanner, 1969). As such, I record GSDs in the phi scale and report errors in the phi scale in the results. The tool requires an input drainage network GIS feature class, for which GSDs are estimated and attributed to each feature. The layer should include attributes representing reach-averaged slope and some measure of flow or a proxy for flow magnitude (e.g., a two-year recurrence interval flow or drainage area; I used cumulative annual precipitation within the basin upstream of each reach), with a value for each feature.

### 2.1 Algorithm

Two different processes can be used, depending on whether one or multiple pebble count measurements are being used for calibration. For best results, multiple measurements spanning a range of stream sizes and gradients should be used. However, data availability is often limited, sometimes to a single measurement for a basin, so the tool also

uses a novel algorithm to estimate GSDs in such circumstances. The algorithm steps are listed below, followed by

more in-depth descriptions.

**Both (single measurement and multiple measurements):**

1) Calculate $D_{16}$, $D_{50}$, and $D_{84}$ and their uncertainty for measurement reach(es) from GSD measurements

(pebble counts).

**Single Measurement:**

2) Find the flow depth required to transport the $D_{16}$, $D_{50}$, and $D_{84}$ at the measurement reach (critical depth).

3) Find hydraulic geometry coefficients associated with the critical depth.

4) Extrapolate critical depth to each drainage network segment based on a parameterized hydraulic

geometry relationship.

5) Estimate $D_{16}$, $D_{50}$, and $D_{84}$ at each segment using the Shields equation and a chosen method for

estimating critical Shields stress.

**Multiple:**

6) Calibrate functions for $D_{16}$, $D_{50}$, and $D_{84}$ that relate grain sizes to the product of slope and flow using all

available measurements.

7) Estimate $D_{16}$, $D_{50}$, and $D_{84}$ at each segment using calibrated empirical function.

**Both:**

8) Fit skew-normal distribution parameters (location, scale, and shape) for each pebble count measurement.

Take the average value of each parameter to use as a starting point in estimating the parameters for non-

measurement reaches.

9) For each drainage network segment, fit new skew-normal distribution parameters using the parameters

from the measurement reaches as a starting point. These parameters are adjusted until the error between

$D_{16}$, $D_{50}$, and $D_{84}$ found from generating a new distribution is minimized compared against the actual values

attributed to the segment in step 5 or 7.

10) Generate a distribution using the solved parameters, removing fractions finer than sand for the segment.

Calculate the fraction of the resulting distribution within each half-phi size interval and store the data in the

output .json file.

### 2.1.1 Calculating $D_{16}$, $D_{50}$, and $D_{84}$ from measurements and quantifying uncertainty

For each pebble count, the data is bootstrapped with replacement 100 times and sampled until the number of values

is 80% of the total number of measurement values. For each bootstrap, the 16[th], 50[th], and 84[th] percentile of the data

is recorded, generating a distribution of each consisting of 100 values. The values for $D_{16}$, $D_{50}$, and $D_{84}$ are then set

as the mean of these distributions with the uncertainty being characterized as plus or minus two standard deviations.

Additionally, the reach slope and flow attribute are recorded (from the drainage network attributes) in order to

parameterize relationships between the product of slope and flow and the different grain sizes.

**2.1.2 Critical Depth**

In the case where a pebble count at a single location is used, after the $D_{16}$, $D_{50}$, and $D_{84}$ are calculated, the depth

required to mobilize each of these sizes (critical depth) is estimated using the Gilbert and Wilcox (2024a) method

for calculating critical Shields stress, along with the Shields function. Critical Shields stress for a given grain size $D_i$

is calculated as


$$\tau_{c,i}^{*} = a \left( \frac{D_i}{D_{50}} \right)^{-0.67}$$
(1)

where the subscript $i$ denotes a given grain size fraction, and the coefficient $a$ varies as a function of the $D_{84}$ to $D_{50}$ ratio,


$$a = 0.029 \; if \; \frac{D_{84}}{D_{50}} \leq 2 \tag{2a}$$

$$a = 0.043 ln \left(\frac{D_{84}}{D_{50}}\right) - 0.0005 \; if \; \frac{D_{84}}{D_{50}} > 2 \tag{2b}$$

The critical Shields number is given by the function


$$\tau_{c,i}^* = \frac{\tau_{c,i}}{(\rho_s - \rho)gD} \tag{3}$$

where $\tau_{c,i}$ is the critical boundary shear stress at which the $i^{th}$ size fraction of the bed becomes mobile. Shear stress is typically estimated using the depth-slope product (which assumes steady, uniform flow),


$$\tau = \rho g h S. \tag{4}$$

In equations 3 and 4, $g$ is acceleration due to gravity (9.81 m/s$^2$), $\rho_s$ is the density of sediment (2650 kg/m$^3$), $\rho$ the density of water (1000 kg/m$^3$), $h$ is average flow depth (m), and $S$ is reach averaged slope (m/m). $\tau$ can be solved for

by multiplying the denominator of the Shields function by $\tau^*$, and critical depth for the size fraction $i$ can then be estimated by rearranging the depth-slope product,

$$h_{c,i} = \frac{\tau_{c,i}}{\rho g S}. \tag{5}$$

### 2.1.3 Hydraulic Geometry Coefficients

In order to extrapolate critical depth values to each drainage network segment, hydraulic geometry coefficients are parameterized. Hydraulic geometry equations scale channel geometry with discharge (Leopold & Maddock, 1953). In downstream hydraulic geometry relationships, an exponent of 0.4 is often used for depth (Morel et al., 2020):

$$h = \alpha Q^{0.4} \tag{6}$$

where $Q$ is discharge. In this case, $Q$ can be substituted for drainage area *or* whatever other flow proxy attribute is being used. $\alpha$ is simply solved for in the measurement reach, and used for every other drainage network segment.

### 2.1.4 Estimating Grain Size (Single Measurement)

After the coefficient in equation 6 is calculated, critical depth for $D_{16}$, $D_{50}$, and $D_{84}$ is estimated for each reach, based on the reach's flow proxy attribute value. Shear stress at that depth is calculated using equation 4 with the calculated critical depth, and then the Shields formula (equation 3) is rearranged to solve for $D$.

$$D_i = \frac{\tau_{c,i}}{(\rho_s - \rho) g \tau_{c,i}*} \tag{7}$$

where the subscript $i$ represents the specific grain size percentile (i.e., 16, 50, 84). Sections 2.1.1 through 2.1.4 are summarized graphically in figure 1,

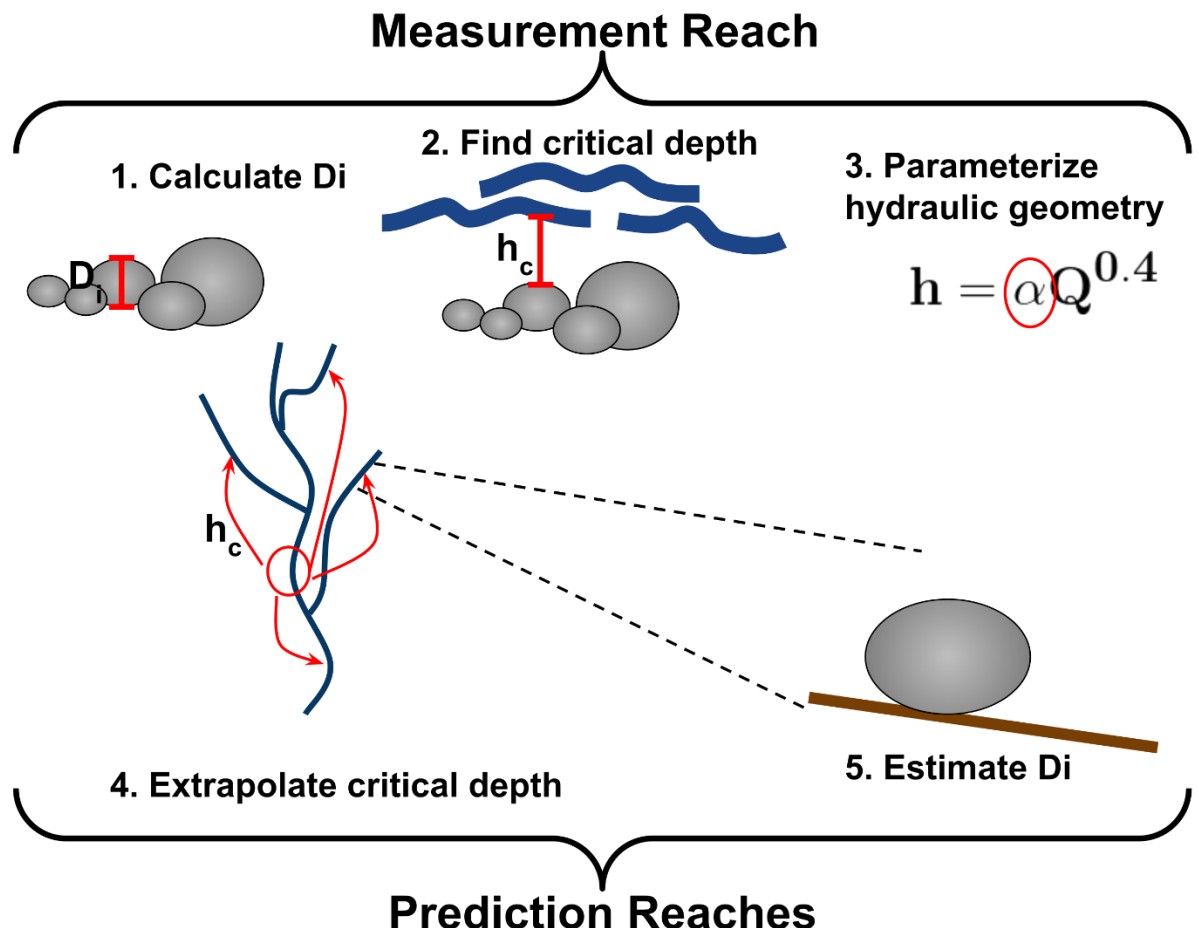

**Measurement Reach**

**1. Calculate Di**

$D_i$

**2. Find critical depth**

$h_c$

**3. Parameterize hydraulic geometry**

$$h = \alpha Q^{0.4}$$

$h_c$

**4. Extrapolate critical depth**

**5. Estimate Di**

**Prediction Reaches**

Figure 1. A graphical representation of the algorithm for predicting grain sizes at all reaches of a drainage network using measurements from a single reach. $D_{16}$, $D_{50}$, and $D_{84}$ are calculated from a pebble count (1) at a measurement reach, the critical depth for that grain diameter estimated (2), and hydraulic geometry coefficients for that depth calculated (3). That coefficient is used to extrapolate critical depth to the remainder of the drainage network (4), and used in a rearrange Shield's Equation to estimate the corresponding grain size at every reach of the drainage network (5).

**2.1.5 Estimating Grain Size (Multiple Measurements)**

When multiple pebble counts are used with the tool, three functions are parameterized to calculate $D_{16}$, $D_{50}$, and $D_{84}$ respectively. For each of the three grain size fractions, a least squares regression is used to fit a relationship predicting grain size as a function of the product of slope and flow (Figure 2). The interaction of these two variables is largely able to explain variation in grain size. When reaches are steeper, they tend to have coarser beds, as flow competence is higher and larger particles can be mobilized. However, a steep headwater stream with little flow will

have a much finer GSD than a higher-order stream with the same slope, where the combination of slope and flow result in much greater flow competence. The values of slope and flow for every other reach in the drainage network are then used to predict $D_{16}$, $D_{50}$, and $D_{84}$ using the fitted relationships.

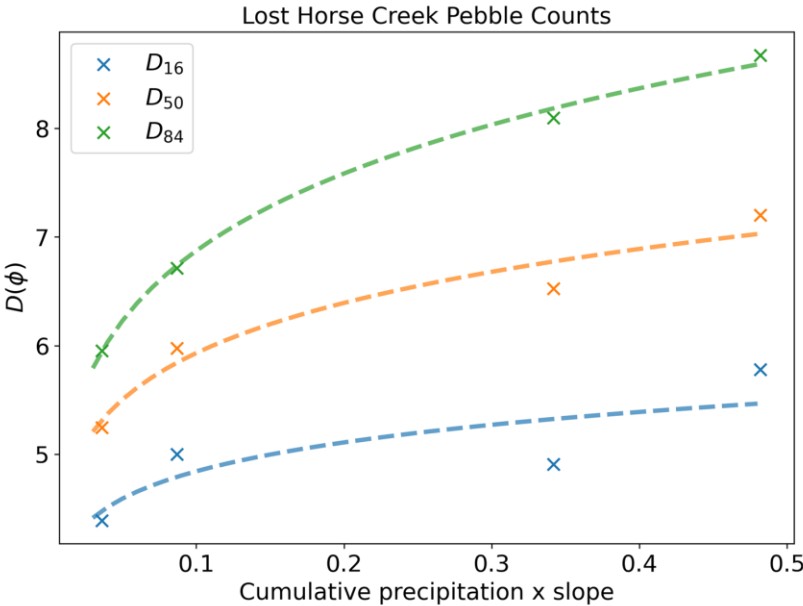

**Figure 2. Parameterized relationships (lines) for calculating $D_{16}$, $D_{50}$, and $D_{84}$ as a function of the product of slope and flow using four pebble counts at different locations within a single basin in the study area. In this application, I used precipitation within the upstream basin as the flow proxy, therefore the x axis is the product of these precipitation values and the reach-averaged slope.**

In the case of both single and multiple measurements, as $D_{16}$, $D_{50}$, and $D_{84}$ are calculated for each reach, the value is stored in the attribute table along with a minimum and maximum value (based on the uncertainty previously calculated following Section 2.1.1). These values are subsequently used to calculate the full GSDs for the reach.

**2.1.6 Estimating Full Grain Size Distributions**

For each grain size measurement, the data is converted to phi values. Typically, phi value distributions for coarse-bedded channels are negatively skewed. The parameters of a best-fit skew-normal distribution (location, scale, and shape) are therefore calculated (Figure 3A). In the case of multiple measurements, the parameters are found for each measurement, and the mean value of each parameter is used as a starting point for estimating parameters at each individual reach.

For each network reach, a distribution is generated using the parameters calculated from the measurement data. An
optimization process is then implemented, varying the parameters until the errors between $D_{16}$, $D_{50}$, and $D_{84}$ of a
distribution generated from the new parameters and the values of $D_{16}$, $D_{50}$, and $D_{84}$ calculated for that segment
previously (Section 2.1.4 or 2.1.5) are minimized (Figure 3B). These optimized parameters are used to generate a
final distribution. Values finer than sand are removed (in this application to coarse bedded rivers). From the
remaining values, the fraction within each half-phi size interval is calculated, and the values stored in a .json file.


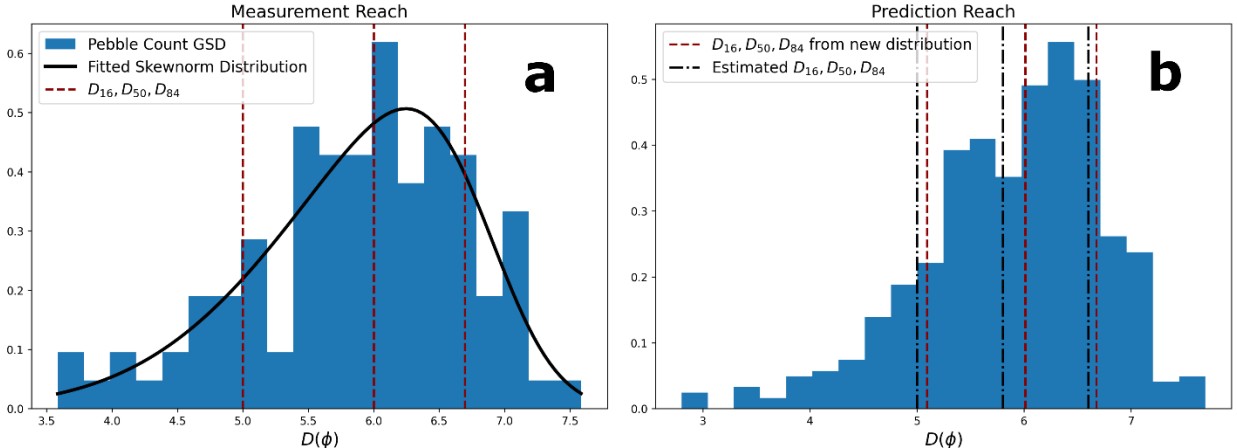

**Figure 3. Pebble count measurements are used to fit a skew-normal distribution and calculate its parameters (A). Those**
**parameters are then used to generate new distributions in non-measurement (prediction) reaches (B). The parameters are**
**then adjusted until the errors between $D_{16}$, $D_{50}$, and $D_{84}$ from newly generated distributions and previously predicted**
**values for $D_{16}$, $D_{50}$, and $D_{84}$ are minimized.**

### 3 Application

To test its effectiveness, I applied the algorithm to two collections of mountain river sub-basins within the larger
Bitterroot River basin of Western Montana (Figure 4). Within each basin, I performed pebble counts at several
locations spatially dispersed across the watershed, capturing varying stream sizes and gradients. One basin in each
collection was used to calibrate the model, after which it was applied to the remaining basins. The first collection

consisted of basins with dendritic networks flowing through v-shaped valleys, where steep hillslopes physically

connected to channels are prevalent throughout the watersheds (Woods, Rye, Sleeping Child, and Burnt Fork

Creeks). The second collection was made up of streams within formerly glaciated mountain basins (Blodgett,

Roaring Lion, and Lost Horse Creeks), with u-shaped valleys and wider valley bottoms, and greater variability in

gradients of hillslopes adjacent to channels as a result of complex erosion patterns and valley geometries common in

post-glacial landscapes (Hassan et al., 2019). Because of the greater uniformity in hillslope gradients adjacent to

channels in first group, I infer generally high hillslope-channel connectivity, and refer to this group hereafter as the

'uniform group.' In the second group, because hillslope gradients adjacent to channels were lower and more

variable, I infer lower and more variable hillslope-channel connectivity, and refer to this group hereafter as the

'variable group.' After running the model using multiple calibration measurements, I ran it for each basin using a

single calibration measurement within the basin, allowing for comparison of results using the two methods.

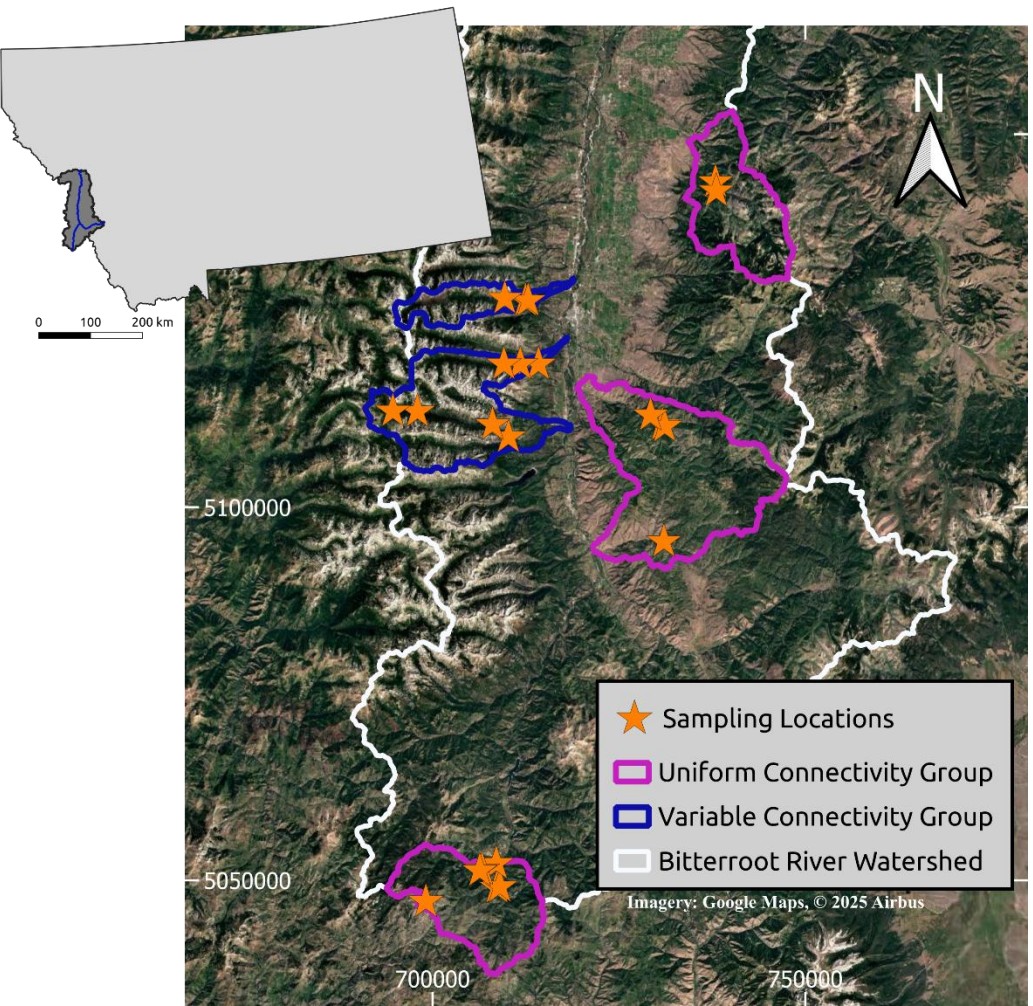

**Figure 4. Locations of grain size sampling within the Bitterroot River basin, and the location of the Bitterroot basin within Montana. Coordinates are in WGS84 UTM 11N. Imagery: Google Maps, © 2025 Airbus.**

### 4 Results

For model results using multiple pebble count measurements for calibration (Table 1), the average discrepancy between predicted and measured size was 6 mm for $D_{16}$, 14 mm for $D_{50}$, and 32 mm for $D_{84}$, or mean absolute percent error (MEAP) of 26.1%, 25.8%, and 26.1% respectively. In comparing predictions to measurements in the phi scale, this equates to MEAP of 7.6% for $D_{16}$, 5.3% for $D_{50}$, and 4.4% for $D_{84}$. All combined, the MEAP for phi values was 5.8% (26 % for mm). With the exception of $D_{84}$, which had similar error in both cases, errors were lower

for the uniform group (6%, 4.1%, and 5% MEAP for phi values) than for the variable group (9.1%, 6.4%, and 3.9%

MEAP for phi values).

Within the uniform group, the geology of three of the four watersheds is primarily metamorphic (gneisses and schists). By contrast, the fourth basin is underlain by sedimentary and metasedimentary (e.g., quartzite) rocks. If the two sites with different geology than the others (BF and GC) are excluded, error is further reduced (5.1%, 1.9%, and 2.5% MEAP for phi values). In the variable group, errors were the highest where GSD was overpredicted in a very low-gradient, unconfined reach.

*Table 1. Grain size estimate results using multiple calibration measurements.*

| | | Measured | | | Predicted | | | Percent Error (mm) | | | Percent Error (phi) | | |
|---|---|---|---|---|---|---|---|---|---|---|---|---|---|
| | Site | D16 | D50 | D84 | D16 | D50 | D84 | D16 | D50 | D84 | D16 | D50 | D84 |
| *Uniform connectivity* | WC1 | 15 | 35 | 73 | 12 | 35 | 76 | 20.0% | 0.0% | 4.1% | 8.2% | 0.0% | 0.9% |
| | WC2 | 24 | 48 | 87 | 16 | 45 | 103 | 33.3% | 6.3% | 18.4% | 12.8% | 1.7% | 3.8% |
| | WC3 | 5 | 15 | 37 | 5 | 15 | 37 | 0.0% | 0.0% | 0.0% | 0.0% | 0.0% | 0.0% |
| | RC | 15 | 50 | 128 | 19 | 57 | 142 | 26.7% | 14.0% | 10.9% | 8.7% | 3.3% | 2.1% |
| | SC1 | 30 | 88 | 208 | 30 | 87 | 213 | 0.0% | 1.1% | 2.4% | 0.0% | 0.3% | 0.4% |
| | SC2 | 35 | 71 | 130 | 30 | 85 | 208 | 14.3% | 19.7% | 60.0% | 4.3% | 4.2% | 9.7% |
| | TB | 21 | 53 | 150 | 22 | 62 | 152 | 4.8% | 17.0% | 1.3% | 1.5% | 4.0% | 0.3% |
| | BF | 39 | 69 | 120 | 26 | 84 | 205 | 33.3% | 21.7% | 70.8% | 11.1% | 4.6% | 11.2% |
| | GC | 14 | 29 | 66 | 17 | 54 | 132 | 21.4% | 86.2% | 100.0% | 7.4% | 18.5% | 16.5% |
| | **Mean Absolute Percent Error** | | | | | | | **17.1%** | **18.4%** | **29.8%** | **6.0%** | **4.1%** | **5.0%** |
| *Va* | LHSF | 30 | 96 | 291 | 40 | 116 | 291 | 33.3% | 20.8% | 0.0% | 8.5% | 4.1% | 0.0% |

| | | | | | | | | | | | | |
|---|---|---|---|---|---|---|---|---|---|---|---|---|
| **LHNF** | | 57 | 150 | 380 | 45 | 137 | 370 | 21.1% | 8.7% | 2.6% | 5.8% | 1.8% | 0.4% |
| **LHO** | | 35 | 65 | 104 | 29 | 61 | 115 | 17.1% | 6.2% | 10.6% | 5.3% | 1.5% | 2.2% |
| **LHP** | | 21 | 39 | 63 | 22 | 40 | 62 | 4.8% | 2.6% | 1.6% | 1.5% | 0.7% | 0.4% |
| **RL** | | 32 | 76 | 184 | 33 | 84 | 190 | 3.1% | 10.5% | 3.3% | 0.9% | 2.3% | 0.6% |
| **RLTX** | | 28 | 83 | 286 | 43 | 148 | 436 | 53.6% | 78.3% | 52.4% | 12.9% | 13.1% | 7.5% |
| **RLA** | | 11 | 22 | 40 | 25 | 50 | 87 | 127.3% | 127.3% | 117.5% | 34.2% | 26.6% | 21.1% |
| **BC** | | 38 | 84 | 172 | 33 | 80 | 178 | 13.2% | 4.8% | 3.5% | 3.9% | 1.1% | 0.7% |
| **BCF** | | 37 | 97 | 304 | 46 | 138 | 396 | 24.3% | 42.3% | 30.3% | 6.0% | 7.7% | 4.6% |
| **BCG** | | 20 | 52 | 137 | 29 | 64 | 129 | 45.0% | 23.1% | 2.8% | 12.4% | 5.3% | 1.2% |
| **Mean Absolute Percent Eroor** | | | | | | | | **34.3%** | **32.4%** | **22.8%** | **9.1%** | **6.4%** | **3.9%** |
| **Mean Absolute Percent Error (all)** | | | | | | | | **26.1%** | **25.8%** | **26.1%** | **7.6%** | **5.3%** | **4.4%** |


Using a single calibration measurement (Table 2), errors were slightly higher than they were using multiple

calibration measurements (11 mm, 21 mm, 48 mm; 13.1%, 6.6%, 5.7%). The average error of all measurements was

8.4%.


*Table 2. Grain size estimate results using single calibration measurements.*

| | | *Measured* | | | *Predicted* | | | *Percent Error (mm)* | | | *Percent Error (phi)* | | |
|---|---|---|---|---|---|---|---|---|---|---|---|---|---|
| | **Site** | **D16** | **D50** | **D84** | **D16** | **D50** | **D84** | **D16** | **D50** | **D84** | **D16** | **D50** | **D84** |
| *Variable connectivity* | **BC** | 38 | 84 | 172 | 37 | 87 | 173 | 2.6% | 3.6% | 0.6% | 0.7% | 0.8% | 0.1% |
| | **BCF** | 37 | 97 | 304 | 87 | 206 | 452 | 135.1% | 112.4% | 48.7% | 23.7% | 16.5% | 6.9% |
| | **BCG** | 20 | 52 | 137 | 25 | 60 | 115 | 25.0% | 15.4% | 16.1% | 7.4% | 3.6% | 3.6% |
| | **LHSF** | 30 | 96 | 291 | 28 | 88 | 270 | 6.7% | 8.3% | 7.2% | 2.0% | 1.9% | 1.3% |
| | **LHNF** | 57 | 150 | 380 | 56 | 174 | 530 | 1.8% | 16.0% | 39.5% | 0.4% | 3.0% | 5.6% |
| | **LHO** | 35 | 65 | 104 | 12 | 48 | 146 | 65.7% | 26.2% | 40.4% | 30.1% | 7.3% | 7.3% |
| | **LHP** | 21 | 39 | 63 | 11 | 33 | 101 | 47.6% | 15.4% | 60.3% | 21.2% | 4.6% | 11.4% |
| | **RL** | 32 | 76 | 184 | 29 | 77 | 184 | 9.4% | 1.3% | 0.0% | 2.8% | 0.3% | 0.0% |
| | **RLTX** | 28 | 83 | 286 | 70 | 184 | 466 | 150.0% | 121.7% | 62.9% | 27.5% | 18.0% | 8.6% |
| | **RLA** | 11 | 22 | 40 | 13 | 33 | 77 | 18.2% | 50.0% | 92.5% | 7.0% | 13.1% | 17.8% |
| *Uniform connectivi* | **WC1** | 15 | 35 | 73 | 16 | 32 | 53 | 6.7% | 8.6% | 27.4% | 2.4% | 2.5% | 7.5% |
| | **WC2** | 24 | 48 | 87 | 23 | 46 | 82 | 4.2% | 4.2% | 5.7% | 1.3% | 1.1% | 1.3% |
| | **WC3** | 5 | 15 | 37 | 13 | 26 | 53 | 160.0% | 73.3% | 43.2% | 59.4% | 20.3% | 10.0% |
| | **SC1** | 30 | 88 | 208 | 36 | 73 | 130 | 20.0% | 17.0% | 37.5% | 5.4% | 4.2% | 8.8% |

| | | | | | | | | | | | | |
|---|---|---|---|---|---|---|---|---|---|---|---|---|
| **SC2** | 35 | 71 | 130 | 36 | 74 | 134 | 2.9% | 4.2% | 3.1% | 0.8% | 1.0% | 0.6% |
| **TB** | 21 | 53 | 150 | 35 | 71 | 145 | 66.7% | 34.0% | 3.3% | 16.8% | 7.4% | 0.7% |
| | | | | | | | | | | | | |
| **Mean Absolute Percent Error (all)** | | | | | | | 45.2% | 32.0% | 30.5% | 13.1% | 6.6% | 5.7% |

One useful feature of the model is that, as previously mentioned it attributes geospatial data with the grain size predictions for $D_{16}$, $D_{50}$, and $D_{84}$. Figure 5 highlights how this allows for identification of spatial patterns in grain size throughout a basin. As qualitative validation of these spatial patterns, obvious differences in grain size distributions of different reaches based on photographs are captured in the model predictions (Figure 6).

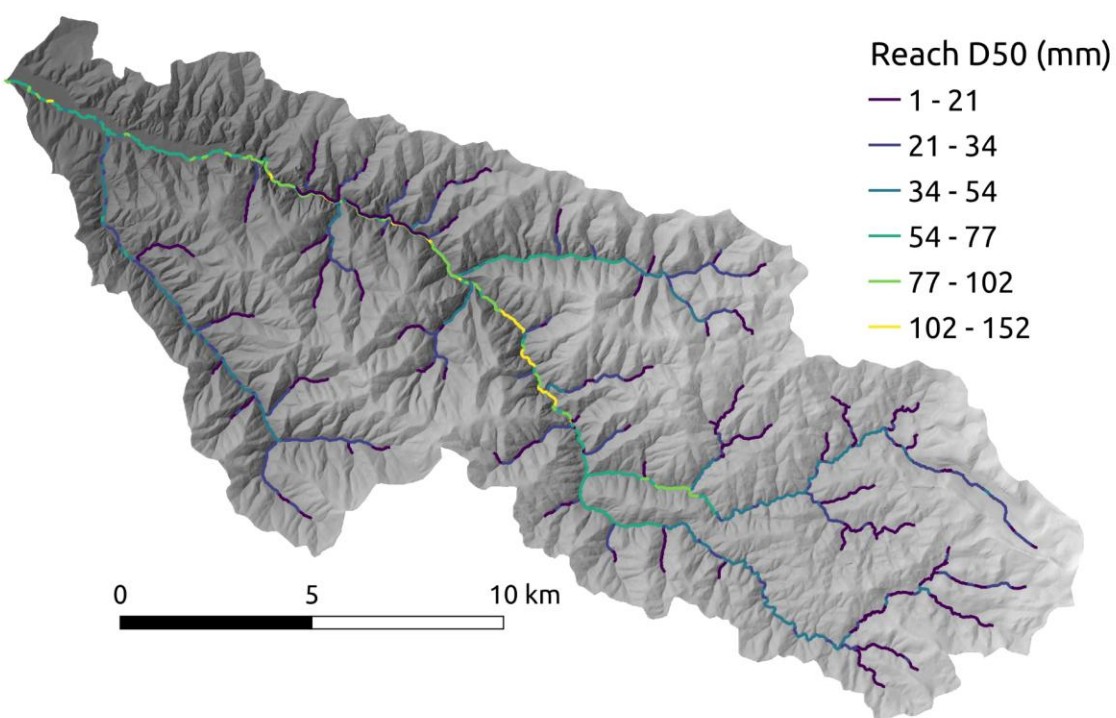

**Figure 5. The drainage network for the Sleeping Child Creek basin, with each reach attributed by its predicted $D_{50}$ value. The network is overlain on a hillshade to show the topography hypothesized to result in uniformly high hill-slope channel connectivity.**

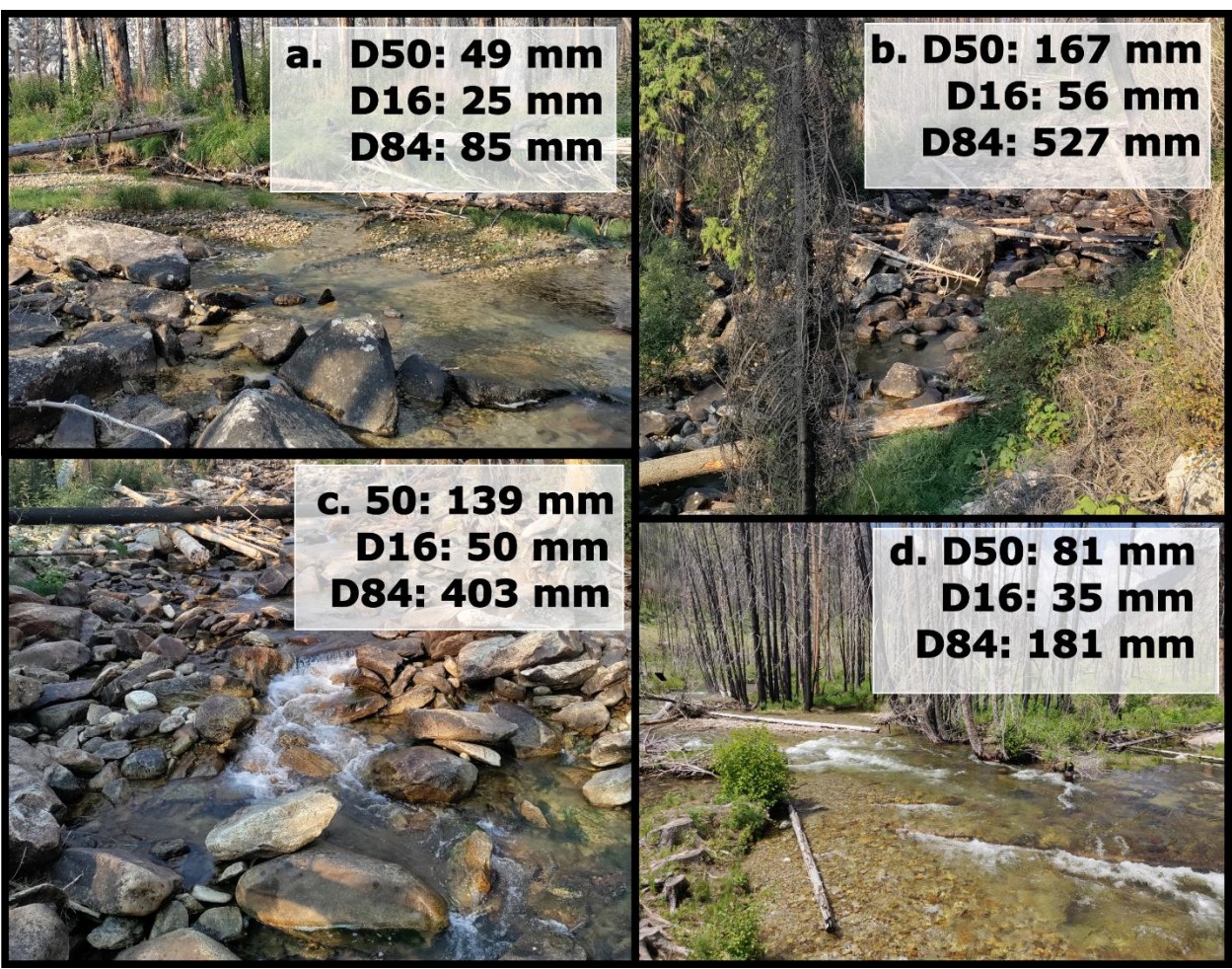

**Figure 6. Photographs along different reaches of Roaring Lion Creek with the predicted D$_{16}$, D$_{50}$, and D$_{84}$ values.**


The .json format is simply a form of text file that is useful for storing information in a human readable format that is also interoperable with various programming languages. The output .json from the tool stores all of the grain size information produced by the tool for each reach. Specifically, this includes the calculated D$_{16}$, D$_{50}$, and D$_{84}$, as well as the fraction of each half-phi size interval in the full GSD.


**5 Discussion and Conclusion**

One of the limitations of the tool is that by using variables related to flow competence to predict grain sizes, it assumes that grain size distributions are hydrologically driven. In mountain drainage basins, as previously mentioned, grain sizes can be driven by, and vary greatly as a function of hillslope-channel connectivity and associated erosion and transport processes. As a result, mountain drainage networks often include channels that are completely non-alluvial, whose bed composition is entirely a function of non-fluvial processes (e.g., glacial erosion, rockfall, or other hillslope sediment production processes). In these cases, where the GSD is entirely a function of hillslope sediment delivery with no fluvial reworking, this model is not applicable for predicting grain sizes. In our study basins, the majority of the sediment delivered from hillslopes to channels in the uniform group watersheds is mobile at some flood flows. As a result, our prediction errors were fairly uniform across all sites. In contrast, much of the sediment delivered from hillslopes to channels in the variable group basins is large enough to be immobile at all flows. As a result, prediction errors in these basins were more variable.

Overall, results from the application of this algorithm presented here suggest that even in mountainous watersheds where GSDs are also a function of hillslope processes, using a hydraulics-based approach for estimating GSDs provides reasonable estimates.

Estimates of GSDs produced using this method may vary based on the input data used and the method of data collection. Grain size values are a function of flow or a flow proxy. Because these empirical relationships are calculated as a function of these data, the specific metric used should not have a large effect on the outputs as long as the data is spatially consistent with similar accuracy across basins. The method used to collect GSD measurements may result in a different distribution shape, which would affect the distribution parameters used to generate GSDs in the prediction reaches. However, results using different methods are typically fairly similar (Strom et al., 2010). In the single measurement method, because a single hydraulic geometry coefficient is calculated from the measurement reach, if the reach is notably different from other reaches in the basin (e.g., much rougher, which could result in greater depth for the discharge value), the coefficient may be a poor fit for much of the basin and result in systematic under or overestimation of grain sizes in the rest of the reaches.

The highest prediction error for these basins occurred in a very low gradient reach where grain sizes were overestimated. This reach, although it contained hillslope derived boulders along channel margins, had a sand and pea gravel bed. This approach is designed for gravel bedded reaches where GSDs typically follow a skew-normal distribution, and this reach was an outlier within the watershed that likely fell outside of these assumptions. The approach could be modified (e.g., use a different distribution for different types of beds), however this highlights the importance of accounting for the assumptions of the tool when interpreting the outputs. In our results, there was no heteroscedasticity in the errors for $D_{16}$, $D_{50}$, and $D_{84}$ calculated as phi values (p=0.13, Breusch-Pagan test), indicating that uncertainty is similar across the full GSD using this approach.

In this application, I used single pebble counts to represent the GSD of entire reaches in the interest of gathering broad spatial coverage. Reaches are better characterized by performing pebble counts within each distinct geomorphic unit present, and subsequently taking the area-weighted mean (Kondolf, 1997). In applications where higher accuracy is desired for a smaller geographic area, calibrating the model with these types of pebble counts may produce better results.

Comparing measured GSDs to aerial imagery for very coarse-bedded reaches suggests that I may have had a bias for the wetted channel at the time of sampling. Consequently, I may have underestimated the size distribution by oversampling the more mobile fraction present near the thalweg relative to other, coarser grains distributed throughout the channel (Figure 7). In very coarse beds, where it is more challenging to walk over the whole bed, a random walk sampling frame is likely less applicable and less effective. In these cases, a different sampling frame (e.g. gridded sampling) could improve sampling of all sizes present in the bed.

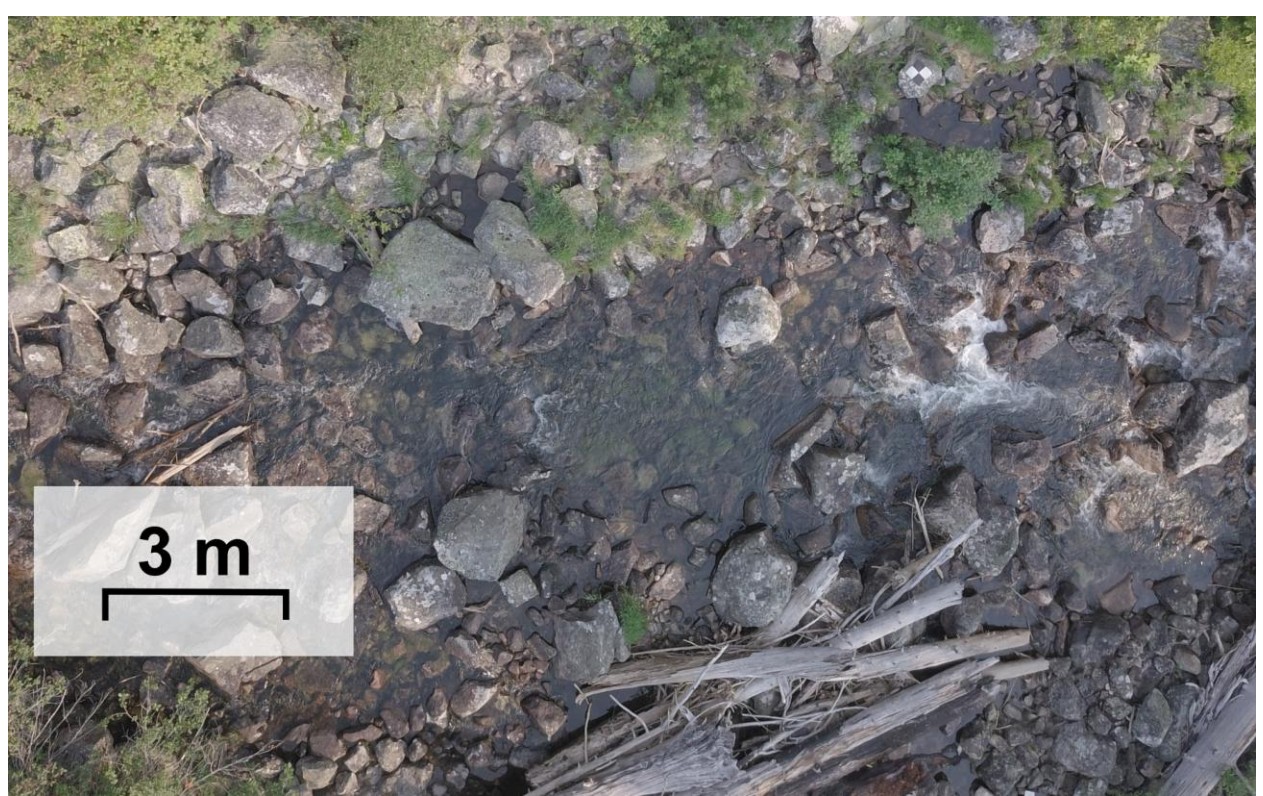

**Figure 7. Aerial imagery of a sample reach of Blodgett Creek with a scale bar. The wetted portions of the channel have a finer GSD than the bed as a whole.**

Within the uniform group, when using a basin with metamorphic lithology for calibration, errors were the greatest in the one sub-basin underlain by different geology. This highlights the control geology plays on GSDs, and

consequently sediment yield, in line with previous findings that geology is a primary control on sediment supply to channels in the Northern Rocky Mountains (Mueller & Pitlick, 2013). While geology exerts control at the landscape scale, other factors can influence variation and GSD at local scales; for example confinement (O'Brien et al., 2019), large wood (Welling et al., 2021; Wohl & Scott, 2017), and beaver dam building activity (Levine & Meyer, 2014). These factors are not included in the model, but should be considered when applying it.


The spatial variability in channel bed grain sizes arises from the interplay between sediment delivery, both from upstream and from adjacent hillslopes, and the competence of the flow to transport that material. These differences

can create large variability in transport rates from reach to reach, driving heterogeneity in fluvial sediment connectivity with implications for landscape evolution, habitat, and human infrastructure. The algorithm presented

here provides an easily applicable method to use pebble counts to extrapolate estimates for full GSDs to an entire drainage network, allowing for increased accuracy in watershed-scale sediment transport estimates.

**Data Availability**

The pebble counts used in this work are available at Gilbert (2024b)

**Code Availability**

grain-size version 1.2.0

Contact: jordan.gilbert@usu.edu

Hardware requirements: PC

Program language: Python

Software required: Minimum processing power required

Program size: 70 KB

The code for applying the algorithm is found at https://github.com/jtgilbert/grain-size (Gilbert 2024c)

**Competing Interests**

The author declares that he has no conflict of interest.

**Acknowledgements**

I thank Andrew Wilcox for advising me in the work that this manuscript was a part of and for reviewing this manuscript, and Jean Dixon, Brendan Murphy, Payton Gardner, and Doug Brinkerhoff for their reviews of this work. I thank James Gearon and Christopher Hackney for their thoughtful reviews of this manuscript. I thank the Bitterroot National Forest, where field work was conducted.

**DGEFinancial Support**

This work was supported by the National Science Foundation (NSF DGE 1633831, NSF EAR 1644619).

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
