# Peer review of "An extrapolation algorithm for estimating river bed grain size"

_EGUsphere, 2025_

## Author Response (AR1)

This is a well-conceived and valuable contribution. The presented algorithm offers an accessible, reproducible method for extrapolating grain size distributions (GSDs) across drainage networks, filling a clear gap in current practice where most models estimate only D50. The integration with GIS and the provision of open code and data enhance the tool's practical value. I commend the author for these contributions and the well-presented Github repo.

That said, I have a few suggestions that would improve clarity, reproducibility, and usability:

1. The paper reports prediction errors as % Phi error. While Phi is a valid log-transformed scale, it may prove somewhat non-intuitive for many readers. I recommend instead reporting errors in standard SI units (mm), possibly after logging the metric values directly if distribution normalization is desired. Additionally, adopting standard error metrics such as mean absolute percentage error (MAPE), root mean square error (RMSE), and perhaps P90 or maximum error would increase accessibility and transparency. The % Phi column in Table 1 essentially functions like MAPE, calling it that would improve clarity.

This point is well taken in terms of intuition for readers. With that said, I believe the phi scale for grain size is the best to use because absolute size difference is less important than relative size difference. To include both, and for the sake of completeness, I updated the tables to present all errors as percent errors in both standard units as well as phi scale, and then mean absolute percent errors (MAPE) for both.

2. Given that the algorithm is the central contribution of the paper, a schematic diagram or flowchart would be highly beneficial. As it stands, readers must follow a relatively dense procedural description scattered over multiple subsections. A visual overview or, at minimum, pseudocode, would make the methodology more traceable and help others implement or adapt the tool.

I added a figure illustrating the algorithm for the single-measurement method as the existing figures serve to illustrate the multi-measurement and full GSD components of the algorithm.

3. The discussion could benefit from addressing a few key interpretive points:

- Error structure: Do model errors appear heteroscedastic (e.g., increasing with D84 or grain size range)? This would have implications for application and confidence bounds.

Errors were homoscedastic and text has been added in the section of the discussion addressing model errors.

- Model boundaries: Are there identifiable thresholds where the model performs poorly (e.g., very low slopes)? Even a soft guideline would help users avoid misapplication.

I added more text in the discussion about possible sources of uncertainty and assumptions that should be met to apply the model or understood to interpret the outputs.

Minor comment:

Please ensure north is indicated on mapview images (Fig. 6) and add lat lon values to the captions of all images of sampling sites.

I added coordinates to all captions of images of field sites.

This manuscript describes an algorithm to estimate grain size distributions across large spatial (basin) scales based on a small, or limited, initial set of observed grain size distribution data. In doing so this manuscript address a big issue in fluvial geomorphology - that of obtaining spatially dispersed grain size estimates. The algortithm is well developed and produces reliable and approriate estimates. As such, this is a valuable contribution. I do, however, have a few recommendations in terms of additions and minor clarifications that would help explain the alogortithm, it uses and limtiations which I feel are necessary to fully help the reader understand and make the most of the presented algortihm.

I believe there should be a more developed discussion of alternative methods of collecting grain size datasets. As it stands, the author discusses Wolman pebble counts and photo sieving, but other techniques such as laboratory sieving of retrieved samples, laser diffraction and determination from TLS surveys are often used as ways of determining GSD. These warrant disucssion for completeness.

I added a paragraph to the introduction to improve the completeness of the review on methods for characterizing grain sizes.

The limitation discussed on Line 100 - 102 (the approach being suited to fully alluvial rivers) is important but is only considered in one sentence before being passed over. Could the author add more clarification of why this is the case and what this means for the test data provided (are the examples used from purely alluvial rivers)?

I added text in this section to clarify the types of streams the approach is best suited for, and revisited it in the first paragraph of the discussion.

A flow diagram of the algorithm steps would be useful to help the reader visualise and understand the workflows - particulalry with the different pathways dependent on the number of input grain size datasets to start with.

I added a figure illustrating the algorithm for the single-measurement method as the existing figures serve to illustrate the multi-measurement and full GSD components of the algorithm.

The author described how the algortihm is integrated with a GIS platform and makes use of a "geospatial drainage network layer" (lines 103 - 104). However, the remainder of the text describes the alogrihm development and this layer is not returned to in the description. Could the author please add more details of this integration, the role this layer plays (if any) in the calculation of the GSDs and how it is propogated through the algorith. Does the choice of drainage network layer impact the subsequent estimates of GSD based on resolution or data source? How is the drainage network layer generated and what format is required for it to work with the algortithm?

I attempted to add text that more explicitly links to the application to the GIS data throughout (e.g., lines 226-228), and now reference the sections that explain how values are calculated that are subsequently attributed to the drainage network feature class.

Leading from this, I think a greater discussion of the sensitivity of the alogrithm outputs to the input variables would be useful for the reader to assess and understand what the alogrithm produces. How sensitive is the output GSDs to the input flow proxies/values used? Does the method of collection of the input observed GSD impact the resulting spatial estimates (i.e. are Wolman counts better than photo sieving as input data, where multiple inputs are used, can they be from different collection techniques or do you need consistency in techniques for robust outputs?).

I added a paragraph to the discussion addressing the effect of different input datasets or measurement techniques, as well as assumptions in model application and how it affects interpretation of outputs.

Small issues:

Line 62: Check spelling, is 'lighology' meant to be 'lithology'?

Fixed

Line 63: Check spelling, grain = grains, o = of.

Fixed

Line 82: Spelling - streams = stream's.

Fixed

Line 83: Check citation- Snelder etl a. should be et al.

Fixed

Figure 1: X-axis needs explaining. What is the cumulative precipitation x slope?

I added additional text in the caption to explain this.